# Dental Implants in People with Osteogenesis Imperfecta: A Systematic Review

**DOI:** 10.3390/ijerph19031563

**Published:** 2022-01-29

**Authors:** Ole Oelerich, Johannes Kleinheinz, Lauren Bohner, Vera Wiesmüller, Marcel Hanisch

**Affiliations:** 1Department for Prosthodontics and Biomaterials, University Hospital Münster, D-48149 Münster, Germany; 2Research Unit Rare Diseases with Orofacial Manifestations, Department of Cranio-Maxillofacial Surgery, University Hospital Münster, D-48149 Münster, Germany; johannes.kleinheinz@ukmuenster.de (J.K.); lauren.bohner@ukmuenster.de (L.B.); 3Department of Operative and Prosthetic Dentistry, Medical University of Innsbruck, A-6020 Innsbruck, Austria; vera.wiesmueller@i-med.ac.at; 4Department of Oral Surgery and Dental Emergency Care, Faculty of Health, Witten/Herdecke University, D-58455 Witten, Germany

**Keywords:** dental implants, osteogenesis imperfecta, systematic review, rare disease

## Abstract

The aim of this systematic review was to answer the question of whether patients with osteogenesis imperfecta can be prosthetically rehabilitated with dental implants. A protocol was prospectively registered in PROSPERO (CRD42021286368). The inclusion criteria were the presence of osteogenesis imperfecta and the use of implants for prosthetic restorations. Cases in which the inclusion criteria were not met were excluded. PubMed, Web of Science, and Scopus were last searched on 22 August 2021. Quality assessment was performed using the Methodological Quality and Synthesis of Case Series and Case Reports tool. The primary outcome was implant survival. Supporting data were analyzed descriptively. Twelve studies were included. Twenty-three patients received a total number of 116 implants, with 5.0 (±3.8) implants placed per patient. The implant survival rate was 94.0% with a mean follow-up of 59.1 months (±36.1). A limitation of this review was the relatively short follow-up time in some of the included studies; therefore, the survival rate may be overestimated. Nevertheless, the available data showed the loss of only seven implants, with two implants lost due to implant fractures not attributable to the patient. With the limitations of this review and based on the available data, dental implants have a high survival rate in patients with osteogenesis imperfecta. Therefore, dental implants may be a viable treatment option for replacing missing teeth. This research was not funded by external resources.

## 1. Introduction

Osteogenesis imperfecta (OI) is a rare genetic disorder characterized by a defect in collagen type I, resulting in bone fragility and connective tissue disjunction [1,2]. The severity of clinical presentation can vary from mild to severe phenotype, depending on the subtype of the disease. In 1979, Sillence et al. classified OI into subtypes I–IV, ranging from mild to lethal based on clinical and radiological features [2,3]. With advances in genetic sequencing, the classification has been updated, and now a total of 20 subtypes (Sillence types I–XX) are known [4]. In general, vascular alteration, hearing loss, and blue sclerae are clinical signs that occur in patients with OI. The combination of poor bone density and quality can lead to skeletal deformities, spontaneous bone fractures, and impaired bone repair [5]. Therefore, bisphosphonate therapy is the treatment of choice for patients with OI [6]. In addition, interdisciplinary dental treatment may be required to correct jaw and dental abnormalities [7,8]. Orofacial manifestations associated with OI include dentinogenesis imperfecta, dental and skeletal malocclusion, and tooth anomalies [7,8,9,10]. 

Since tooth agenesis is commonly diagnosed in patients with OI (ranging from 10 to 22% in the literature [11,12,13,14]), dental implants may be a reliable treatment option to replace absent teeth. While dental implants are a safe therapy with high survival rates in healthy patients [15], the treatment outcome in patients with OI is questionable due to the complications associated with the disease and its treatment. Not only bone fragility but also bisphosphonate therapy may influence implant treatment [8].

Case reports in the literature suggest that dental implant therapy may be a reliable option for patients with OI [7,8,9,16,17,18,19,20,21,22,23,24]. Oral rehabilitation treatments, ranging from single crowns to implant-supported prostheses, showed an acceptable outcome at short-term follow-up. However, the literature is still sparse and long-term outcomes are still unpredictable. Therefore, it is still unknown whether dental implants can be safely recommended for compromised patients affected by OI.

To evaluate the outcomes of dental implant therapy in patients with OI, this systematic review updated the current scientific evidence to answer the following focused question: Can patients with osteogenesis imperfecta be successfully treated and prosthetically rehabilitated with dental implants?

## 2. Materials and Methods

### 2.1. Protocol Registration

This systematic review reports in accordance with the Preferred Reporting Items for Systematic Reviews and Meta-Analyses (PRISMA) statement [25] (Appendix A). The review protocol was prospectively registered in PROSPERO: International Prospective Register of Systematic Reviews (CRD42021286368).

### 2.2. Search Strategy 

A systematic search was carried out in three different databases, namely PubMed, Web of Science, and Scopus, in order to identify relevant studies. The search query and Boolean operators were based on the “Population Intervention Comparison Outcome” (PICO) (Table 1).

The keywords were used for each database, adapting them to the syntax rules of each database. To maximize the number of possible results, the search query only consisted of the population and the intervention. In addition, a manual search was performed by screening the references of the included studies. All databases were last searched on 22 August 2021. Rayyan software [26] was used to search for duplicates. Individual studies were then screened for inclusion and exclusion criteria. 

### 2.3. Inclusion and Exclusion Criteria

All clinical study designs written in English or German were included in the review. The inclusion criteria were clinical studies, both prospective and retrospective, in which patients diagnosed with osteogenesis imperfecta were prosthetically rehabilitated with dental implants. All studies in which at least one of the placed implants was loaded were included, regardless of the type of prosthetic abutment used. To determine implant survival, at least one follow-up visit after loading was required. Cases in which the inclusion criteria were not met or the full text was not available resulted in exclusion.

### 2.4. Selection Process and Data Extraction

For the initial assessment, two independent reviewers checked the titles and abstracts of each article (O.O. and M.H.) for inclusion or exclusion. Any disagreements were solved by discussion. The interrater reliability had a Cohen’s kappa of 0.855. Quality assessment of the included articles was performed independently by the same two reviewers. Data were independently extracted by a reviewer (O.O.) and entered into a prepared Excel spreadsheet (Excel, Microsoft Corporation, Redmond, Washington, USA; retrieved from https://office.microsoft.com/excel (accessed on 22 August 2021)). Because one of the reviewers (M.H.) is an author of one of the included studies, another reviewer (L.B.) performed the quality assessment for that study [7]. If a study was a follow-up to a previous study, only the most recent data of the same patients were used. Any disagreements were resolved by discussion. Extracted data included: 

Patient characteristics:○Age of patient;○Sex of patient;○Subtype of OI;○If the patient was taking bisphosphonates; ○If the patient was smoking.

Surgical procedures:○If bone augmentation was performed (ridge augmentation or sinus floor elevation);○If antibiotics were administered during surgery;○One- or two-phase surgical procedure.

Implant characteristics:○Number of implants;○Implant position;○Type of implant;○Bone- or tissue-level implant;○Months between placement and loading of implants;○Type of suprastructure on abutment.

Outcome and follow-up○Time of follow-up;○Probing depth;○Radiological bone loss (intraoral single-tooth radiograph or panoramic radiography);○If the implant was lost (with reason of failure);○Time between surgery and failure. 

The main outcomes for evaluating implant survival were (1) the number of failed implants with reason for failure and (2) follow-up time with probing depths and radiographic bone loss. The mean and standard deviation (with range) were reported for synthesis of results. The remaining results were evaluated for descriptive analysis, and proportions were used.

### 2.5. Quality Assessment

For case reports and case series, the tool proposed by Murad et al. (Methodological Quality and Synthesis of Case Series and Case Reports) [27] was used to assess the methodological quality of the included studies (Appendix A). The study from Jensen et al. [19] was also assessed using this instrument, although it included both retrospective and prospective patients, as the retrospective group corresponds to the description of a case series.

The tool consists of eight leading explanatory questions, regarding selection, ascertainment, causality, and reporting. Questions 4, 5, and 6 are mostly relevant to cases of adverse drug events, but it was decided to include Question 4 (Were other alternative causes that may explain the observation ruled out?) anyway because it was considered important for our evaluation, as alternative causes could be an explanation for implant loss. To assess the methodological quality of each study, Murad et al. proposed making an overall judgment on the questions considered most critical in the specific clinical scenario. In this review, it was decided to divide the overall assessment into three categories (good quality of reporting, medium quality of reporting, and low quality of reporting) and include all studies with either good or medium quality of reporting for the final data extraction. It was decided that studies providing enough information to treat a similar patient would be classified as “good quality of reporting”, studies in which some information was missing or incompletely presented but sufficient to treat a similar patient would be classified as “medium quality of reporting”, and studies that did not provide enough information to treat a similar patient would be classified as “low quality of reporting”.

## 3. Results

### 3.1. Literature Search

The database search strategy yielded 65 results. After excluding duplicates, the titles and abstracts of 39 records were checked for inclusion criteria by the reviewers. Twelve articles were selected for quality assessment and full-text screening and were included in the systematic review (Figure 1).

### 3.2. Study Characteristics

The included studies were published between 2000 and 2021. Twelve papers [7,8,9,16,17,18,19,20,21,22,23,24] were included. All but one of the included studies were either single case reports or case series. The number of subjects ranged from 1 to 13, and the number of implants placed ranged from 2 to 46 implants. (Table 2). The study from Jensen et al. [19] included both retrospective patients (which can be considered a case series) and a prospective group. The study from Myint et al. [9] was a follow-up study of the prospective group previously studied by Jensen et al. For the specific patients included in these two studies, the most recent data were used for further analysis. 

### 3.3. Quality Assessment

Nine studies [7,8,9,16,19,20,21,22,24] were rated as having good quality of reporting. The remaining three studies [17,18,23] were rated with a medium quality of reporting. An overview of the results of the individual studies can be found in Table 3 with the final evaluation of each study. Because all but one of the studies (Jensen et al.) were single case reports of patients who had sought consultation, there was no evidence that other patients with similar presentation had not been reported. The prospective study by Jensen et al. included clear inclusion and exclusion criteria. It was considered important to rule out alternative causes (Question 4) only if an implant had been lost and there was evidence that the loss of the implant might not have been correlated with the diagnosis of OI. The follow-up time (Question 5) may be too short in most studies to assess final implant survival. Detailed information on the quality assessment of each individual study, with rationale for judgment, is provided in Appendix A.

### 3.4. Description of the Included Studies and Patients 

Table 4 shows the demographic and clinical characteristics of patients with osteogenesis imperfecta rehabilitated with dental implants described in the literature. There were 23 patients (12 females and 11 males) whose mean (±SD) age at the time of treatment was 49.6 (±15.0) years (range 20–75). 

Ten patients (43.5%) were diagnosed with OI type I according to Sillence clinical classification. Four patients (17.3%) were diagnosed with type III OI and six patients (26.1%) were diagnosed with type IV OI. In three cases (13.0%), information on the subtype was missing.

Of the 23 patients, 7 (30.4%) were smokers. Seven patients (30.4%) were described as nonsmokers, whereas information was missing for the remaining nine patients (39.1%).

### 3.5. Bisphosphonates

A total of four patients (17.4%) had taken bisphosphonates in the past. Based on detailed patient medication information, one patient was not taking bisphosphonates, while information was missing for the remaining 18 patients (78.3%). 

One patient received 5 mg zoledronic acid (Aclasta) intravenously every six months. Another patient had received alendronic acid per os for many years. Two years prior to the treatment, the patient’s therapy was replaced with denosumab injections. One other patient took bisphosphonates for some time after the initial study. The patient developed peri-implantitis, but after a surgery to clean the implant surface, the bone regenerated. A different patient received two doses of zoledronate acid, 10 and 4 months before surgery. None of the implants in these four patients failed during the follow-up period.

### 3.6. Bone Augmentation

#### 3.6.1. Ridge Augmentation

For 30 implants (25.8%), the use of ridge augmentation before or during implant placement was reported. Information was missing for 60 implants (51.7%), and 26 implants (22.4%) were placed without bone augmentation. 

Autogenous bone was used for augmentation in 22 implants (18.9%). In 17 of the implants placed (14.6%), bone was harvested from the iliac crest, with demineralized deep-fried bone allograft (Dembone, Pacific Coast Tissue Bank, Los Angeles, CA, USA) and OsteoGraf/N (CeraMed Dental, Lakewood, CO, USA) added in 2 implants and platelet-rich plasma in 4 implants. Autogenous bone was harvested from the mandibular ramus prior to placement of five implants (4.3%), with OsteoGraf/N added for three implants. 

Allografts were used exclusively before eight implants (6.9%).

Autogenous bone from the iliac crest was used in addition to Dembone prior to implantation of one failed implant. 

There was only one patient in whom ridge augmentation was performed and who was taking bisphosphonates. Autogenous bone from the iliac crest and platelet-rich plasma were used and no complications were reported.

#### 3.6.2. Sinus Floor Elevation

A sinus floor elevation was performed before 19 implants (28.8%) were placed into the maxilla. In 55 placed implants (47.4%) it was either stated that no sinus floor elevation was performed or that the implants were inserted into the mandible. In 42 placed implants (36.2%), the information was missing. One of the nineteen implants (5.3%) placed after sinus floor elevation was performed failed during the follow-up period.

### 3.7. Antibiotics

Five of the twenty-three patients (21.7%) received antibiotics before or during implantation. The remaining 18 patients (78.3%) lacked information on antibiotic use. The most frequently used antibiotic was clindamycin, which was used in three patients. One patient received 150 mg of clindamycin every 8 h for 30 days. Another patient received 600 mg of clindamycin every 8 h for 7 days. A different patient also received 600 mg of clindamycin every 8 h during surgery for eight implants, but for an unspecified period. For three of his implants, he received 600 mg of Clindamycin every 8 h for 6 days, and for the remaining five implants, he received 1 g of Kefzol (cefazolin) intravenously preoperatively and 500 mg of Keflex (cefalexin) every 6 h for 10 days.

One patient received 500 mg amoxicillin every 8 h, 1 day before surgery and then for 7 days.

One other patient received cefuroxime at an unspecified dosage and for an unspecified period of time.

### 3.8. Implant Characteristics

The most frequently used implants were the Astra Tech Osseospeed (Dentsply Sirona, Karlsruhe, Germany) (24 implants; 20.7%; 3.5 × 9 mm–5.0 × 11 mm, smallest and largest implant placed; information on all various sizes can be found in Table 4) and Astra Tech Tioblast (17 implants; 14.7%; 3.5 × 11 mm–4.5 × 13 mm). Brånemark System MK III Ti-Unite implants (Nobel Biocare, Kloten, Sweden) were also used 17 times (14.7%; 3.75 × 10 mm–4.0 × 15 mm), and 10 Brånemark bone-tapped implants (Nobel Biocare, Kloten, Sweden) (8.6%; 13–15 mm) were used without more detailed designation. There were 10 NobelActive implants (Nobel Biocare, Kloten, Sweden) (8.6%; 3.5 × 10 mm–5.0 × 10 mm), three NobelDirect implants (Nobel Biocare, Kloten, Sweden) (2.6%; 3.0 × 10 mm–3.0 × 15 mm), and one Nobel (Nobel Biocare, Kloten, Sweden) implant (0.9%) without exact designation. The designations of the other implants are Straumann Standard (Straumann, Freiburg, Germany) (five implants; 4.3%), Straumann Standard Plus SLActive (Straumann, Freiburg, Germany) 3.3 × 10 mm (two implants; 1.7%), Straumann (Straumann, Freiburg, Germany) (two implants; 1.7%; 4.1 × 8 mm–4.1 × 12 mm), MIS C1 (MIS Implants Technologies Ltd., Galilee, Israel) (three implants; 2.6%; 3.75 × 10 mm–3.75 × 11.5 mm), Biomet 3i (Zimmer Biomet, Warsaw, IN, USA) tapered 3.25 × 11 mm (three implants; 2.6%), Paragon Screw-vent (Paragon, Bergenfield, NJ, USA) internal hexed (two implants; 1,7%), and eight implants (6.9%) with only diameter and length without exact designation (3.5 × 10 mm–4.0 × 15 mm). There was no information on the type of implants used in the remaining eight cases (6.9%).

Two Astra Tech Osseospeed implants failed, one before loading (3.5 × 9 mm) and one due to an implant neck fracture (4.5 × 11 mm). One NobelDirect implant (3.0 × 10 mm) and one Brånemark bone-tapped implant (13 mm) also failed before loading. The other implant lost due to implant fracture was an undesignated 3.5 × 13 mm implant, and no information was available on the remaining two lost implants.

### 3.9. Implant Placement and Prosthodontic Rehabilitation

In total, the patients received 116 implants, with a mean (±SD) number of 5.0 (±3.8) implants per patient (range 1–16). Seven patients (30.4%) received implants solely in the maxilla, eight patients (34.8%) received implants solely in the mandible, and eight patients (34.8%) received implants in both the mandible and the maxilla. Sixty-six implants (56.9%) were placed in the maxilla, and 50 implants (43.1%) were placed in the mandible (Table 5). Fifty-seven implants (49.1%) were placed at bone level in 10 patients, 11 implants (9.5%) were placed at tissue level in three patients, and the information was missing for 10 patients and 48 implants (41.4%). In 18 patients, implants were placed in a two-stage procedure in which implant placement and loading were separated in time. In 100 implants (86.2%), prosthetic rehabilitation was performed in this way. In four patients, 14 implants (12.1%) were immediately loaded. Two of those patients only received immediately loaded implants, while the other two received a mixture of immediately loaded implants and implants placed in a two-stage procedure. In two patients, information on only two implants was missing. None of the immediately loaded implants were lost during follow-up. The mean time between implant placement and loading was 8.9 months (±6.3) and ranged from 0 months for immediately loaded implants to 40 months. 

Of all 116 implants, 113 (97.4%) were used for prosthetic rehabilitation. Eight patients were restored with complete or partial overdentures (51 implants, 44.0%). Four of these patients were restored with ridge constructions (32 implants, 27.6%), three with crowns (17 implants, 14.7%), and one patient with locators (two implants, 1.7%) as abutments on the implants. Seven patients (31 implants, 26.7%) were restored with bridges, 10 patients (24 implants, 20.7%) were restored with single crowns, and information on prosthetic treatment was missing for seven implants (6.0%).

### 3.10. Outcome and Follow-Up

Regarding implant survival, only seven implants were lost, giving a survival rate of 94.0% (109 implants). The seven implants were lost in a total of four different patients. Implant failure occurred on average 39.1 months (±29.4; range 3–78 months) after surgery. Three implants (2.6%) failed before loading, and two implant fractures (1.7%) were described (Table 6).

The mean follow-up time was 59.1 months (±36.1) and ranged from 11 to a maximum of 135 months. Most studies reported the extent of radiographic bone loss at follow-up and only two studies reported clinical probing depth around the implants. Radiographic bone loss ranged from 0 to 7 mm with a mean loss of 0.8 mm (±1.3). Probing depths ranged from 2 to 4 mm.

Since the group of patients was very heterogeneous, the survival rates were also analyzed in different subgroups. The subgroups are based on biological characteristics and different treatment procedures (Table 7).

## 4. Discussion

Information on the use of dental implants for the rehabilitation of patients with OI imperfecta is still insufficient. The aim of this systematic review was to summarize the data published in the literature on patients with OI rehabilitated with dental implants so that clinicians can make better treatment decisions and improve the oral health-related quality of life (OHRQoL) of affected patients. The severity of clinical presentation can vary from mild to severe phenotype depending on the subtype of the disease.

Common dental aberrations in patients with OI include malocclusion, with a class III occlusion being the most common [10,12,29,30]; a high prevalence of tooth agenesis [11,12,13,14]; dentinogenesis imperfecta; denticles; and obliteration within the pulp cavity [31]. Those clinical findings can result in reduced OHRQoL [32,33,34]. Due to the increased prevalence of missing teeth, which may be further exacerbated by the involvement of dentinogenesis imperfecta, implants may provide an option to replace the lost teeth, with a survival rate of 94.0% in this present study.

All but one of the studies included in our review were single case reports, and most of them contained detailed information on surgical procedures, type and position of implants, and follow-up. The importance of case reports should not be underestimated, as they contribute to the identification, description, or development of treatments, especially in rare diseases where there are usually not enough patients to conduct robust studies [35,36]. Nevertheless, case reports always carry the risk of incomplete records and missing information for data synthesis. Further studies should focus on detailed reporting of all variables mentioned in this review (e.g., type of OI, smoker, bisphosphonates, antibiotic) to allow for better statistical analysis.

Reviews of the literature were provided in three of the included studies [17,19,21]. However, these reviews were limited to an overview of the case reports to date without going into more detail about the individual results of the studies or the statistical analysis. Therefore, the present study aimed to update the current state of the literature and evidence and to statistically analyze the available data of existing studies.

For the treatment of OI, bisphosphonates are often prescribed to increase bone mineral density in children and adults [6,37,38]. The various influences that bisphosphonates can have on dentistry are not yet fully understood. It is known that bisphosphonates can prevent bone resorption through their direct effect on osteoclasts [39]. However, this desired effect in OI may have further implications. Studies have shown that orthodontic tooth movement was slower in patients treated with bisphosphonates [40,41]. Despite this, affected patients were able to undergo both treatment with orthodontics and orthognathic surgery and achieved stable results. It should be taken into account that patients receiving bisphosphonate therapy have longer orthodontic treatment times and require greater forces [42]. This is especially important for preprosthetic restoration in order to prepare patients for definitive prosthetic treatment.

In the present study, most case reports did not indicate whether the patient was taking bisphosphonates. This information is essential to draw conclusions about the effects of bisphosphonates on implant survival or the occurrence of bisphosphonate-related osteonecrosis of the jaw (BRONJ). Therefore, no conclusion could be drawn in the present study. However, a recent study has shown that patients with a history of bisphosphonates are at no higher risk of implant failure than patients without a history of bisphosphonates [43]. Whether this is also the case in patients with OI and a history of bisphosphonates requires further investigation.

Another study by Contaldo et al. [44] showed that BRONJ does not occur in the pediatric OI population after dental procedures, but it is still unclear whether BRONJ can occur later in life when comorbidities develop. No cases of BRONJ were described in the present study.

Although sinus floor elevation and ridge augmentation are now standardized procedures when insufficient bone is available prior to implant placement, the present study provided little information on these procedures in an OI population [45,46,47,48]. Sinus elevation was performed in 19 implants prior to implant placement and ridge augmentation was reported for 30 implants. Only one implant failed in patient 23, in whom a sinus lift was performed with autogenous bone from the iliac crest in combination with allograft. No implant loss was reported in all other implants placed in which either a sinus floor elevation or ridge augmentation was performed prior to the placement.

There is still no standardized protocol for the preoperative use of antibiotics to prevent implant failure. While Park et al. [49] concluded that practitioners should not routinely use antibiotics in healthy patients, a recent study from Kim et al. [50] described a 53% reduced risk of implant failure in patients who received antibiotics preoperatively. When antibiotics are prescribed prior to dental implant placement, oral amoxicillin is used in most cases, according to a recent study in the United Kingdom [51].

The available data in the present review showed that the use of antibiotics before implant surgery was reported in only five patients. Clindamycin was used in three patients, with a dosage of 600 mg every 8 h being the most common. Other antibiotics used were amoxicillin, cefuroxime, cefalexin, and cefazolin, all of which have been reported in the literature for preoperative treatment of dental implants [52]. 

Again, because of the lack of data in the present study, no conclusions could be drawn about preoperative antibiotic use and implant survival in patients with OI. Detailed information on antibiotic prescribing should be reported in further studies related to implants in patients with OI. However, as suggested by the above-mentioned study, preoperative antibiotic use may reduce the risk of early implant failure.

In order to rehabilitate patients with missing teeth as quickly as possible, immediate loading with predictable results is increasingly performed, both in partially edentulous and completely edentulous patients [53,54,55]. In the present study, 14 implants were immediately loaded, and none of the implants failed during follow-up. In the included study by Prabhu et al. [21], the patient was prosthetically rehabilitated with a total of 10 immediately loaded maxillary implants. The follow-up period for the first implant placed in this study was 48 months, with probing depths ranging from 2 to 4 mm.

Although data were still insufficient to provide predictive results, the included case reports showed that implant treatment with immediate loading protocol may also be a viable option for treating patients with OI.

The implant survival rate was 94,0%. Seven implants were lost due to either biological complications or mechanical complications. Of these seven implants, two were lost due to mechanical complications. Since the purpose of this systematic review was to investigate whether implants are a viable treatment option for patients with OI, particular emphasis should be placed on the implants that were lost due to biological complications. Mechanical complications can occur with any implant but are unrelated to the disease and are caused by treatment errors or material failure. In our study, the implant survival rate without mechanical complications was 95.7%. Two implants lacked a reason for failure, and the remaining three implant failures were detected before loading. 

Follow-up time varied widely, with a mean of just under five years, so the failure rate is likely underestimated because longer follow-up times lead to an increase in the failure rate. Nevertheless, the implant survival rate in patients with OI is only slightly lower than the success and survival rate of implants in daily dental practice (94.0% in patients with OI vs. 98.6% in the general population) [56]. The slightly lower survival rate may be related to the poor bone density and quality associated with the defect in collagen type I in patients with osteogenesis imperfecta.

Incomplete data were the major limitation of this systematic review. Although all studies provided accurate information on implant survival rates and in almost all cases either radiographic bone loss or measurement depths were reported, only two studies [9,19] reported accurately how radiographic bone loss was measured (periapical radiographs in parallel technique were used to assess the bone level visually and compared the situation at follow-up with baseline). Additional information about the patients was missing in all studies. Therefore, it is difficult to draw conclusions about possible alternative causes of implant loss and to compare the studies. Another limitation of this study is the sometimes quite short follow-up time of the individual studies. Although some studies, such as that of Prabhu et al. [24], showed a good survival rate for more than 8 years, in other studies the follow-up time of only 1–2 years is too short to make proper conclusions about implant survival. For this reason, the implant survival rate might be overestimated. Therefore, more long-term studies or follow-up studies, such as that of Myint et al. [9], to the existing case reports are needed. 

It is difficult to make a precise assessment of the implant survival rate as the patients in this review had very heterogeneous treatment procedures and different biological characteristics. It should be considered that the surgical procedures ranged from basic implant surgery to major bone augmentation prior to implant placement, and the prosthetic restoration ranged from single-implant crowns to overdentures on ridge constructions. Therefore, survival rates may vary widely from patient to patient. All cases of implants that have been placed for prosthetic rehabilitation in patients with OI were considered in this review, without narrowing down to a specific subtype, a defined surgical protocol, or a specific prosthetic restoration. Table 7 compares the survival rates of the different treatment procedures in this review, but further controlled studies and case reports are needed to make a more realistic comparison between patients with similar treatment procedures and biological characteristics. Nevertheless, this systematic review provides important data on the possibility of implant placement in patients with OI. Although studies on rare diseases always face the issue of not having enough patients, this review shows a high survival rate for a total of 116 implants placed. The strengths of this systematic review are the prior recording of a protocol, the selection of the best available evidence, and the quality assessment of all included studies.

## 5. Conclusions

Considering the strengths and limitations of this systematic review, prosthetic rehabilitation of patients with osteogenesis imperfecta with dental implants has a high implant survival rate. Therefore, dental implants should be a method of choice when restoring affected patients.

More cases with longer follow-up periods are needed to further evaluate implant survival. It is important that future studies provide complete patient data so that conclusions can be drawn about possible adverse effects or alternative causes of implant failure.

## Figures and Tables

**Figure 1 ijerph-19-01563-f001:**
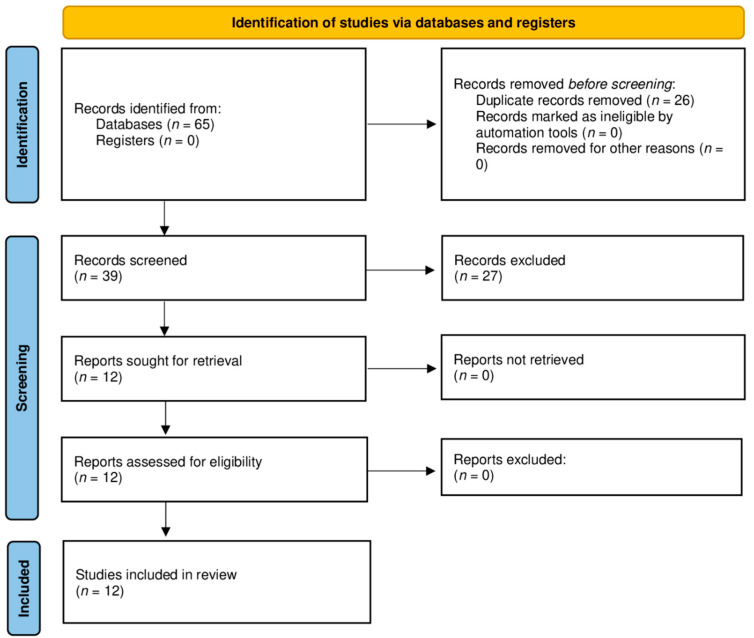
PRISMA 2020 flow diagram for new systematic reviews which include searches of databases and registers only. From: Page, M.J.; McKenzie, J.E.; Bossuyt, P.M.; Boutron, I.; Hoffmann, T.C.; Mulrow, C.D., et al. The PRISMA 2020 statement: an updated guideline for reporting systematic reviews. *BMJ* 2021; 372: n71 [28].

**Table 1 ijerph-19-01563-t001:** Focused question using the PICO approach and search query for the databases.

Focused Question	
PICO model	Can people with osteogenesis imperfecta (P) be successfully treated and prosthetically restored (O) with dental implants (I)?
Search query	#1 (osteogenesis imperfecta) OR (brittle bone disease)
	#2 (dental implants) OR (guided tissue regeneration) OR (alveolar bone grafting)
	#1 AND #2
Study design	All clinical study designs

**Table 2 ijerph-19-01563-t002:** Characteristics of the included studies.

Authors (Year)	Type of Study ^1^	No. of Subjects	No. of Implants
Binger et al. (2006)	CS	1	5
Caicedo-Rubio et al. (2017)	CS	1	3
Friberg et al. (2013)	CS	1	6
Hanisch et al. (2021)	CS	1	2
Jensen et al. (2011) (Myint et al. 2019)	R/P	13	46
Lee et al. (2003)	CS	1	2
Payne et al. (2008)	CS	1	11
Prabhu et al. (2007)	CS	1	11
Prabhu et al. (2018)	CS	1	10
Wannfors et al. (2009)	CS	1	4
Zola et al. (2000)	CS	1	16

^1^ CS—case study/case series; R/P—retrospective/prospective study.

**Table 3 ijerph-19-01563-t003:** Results of the quality assessment using the tool for Methodological Quality and Synthesis of Case Series and Case Reports.

Study	Question 1	Question 2	Question 3	Question 4	Question 5	Question 8	Overall Judgement
Binger et al. (2006)	Yes	Yes	Yes	No	Yes	Yes	Good quality
Caicedo-Rubio et al. (2017)	Yes	Yes	Yes	Yes	Yes	Yes	Good quality
Friberg et al. (2013)	Yes	Partially	Partially	Yes	Partially	Partially	Medium quality
Hanisch et al. (2021)	Yes	Yes	Yes	Yes	Partially	Yes	Good quality
Jensen et al. (2011)	Yes	Yes	Yes	Partially	Yes	Yes	Good quality
Lee et al. (2003)	Yes	Yes	Partially	No	Partially	Yes	Medium quality
Myint et al. (2019)	Yes	Yes	Yes	Yes	Yes	Yes	Good quality
Payne et al. (2008)	Yes	Yes	Yes	No	Partially	Yes	Good quality
Prahbu et al. (2007)	Yes	Yes	Yes	No	Yes	Yes	Good quality
Prahbu et al. (2018)	Yes	Yes	Partially	No	Partially	Yes	Good quality
Wannfors et al. (2009)	Yes	Yes	Partially	No	Partially	Yes	Good quality
Zola et al. (2000)	Yes	Yes	No	No	Partially	Yes	Medium quality

All questions could be answered with Yes, No, or Partially. Question 1: Does the patient(s) represent(s) the whole experience of the investigator (center) or is the selection method unclear to the extent that other patients with similar presentation may not have been reported? Question 2: Was the exposure adequately ascertained? Question 3: Was the outcome adequately ascertained? Question 4: Were other alternative causes that may explain the observation ruled out? Question 5: Was follow-up long enough for outcomes to occur? Question 8: Is the case(s) described with sufficient details to allow other investigators to replicate the research or to allow practitioners make inferences related to their own practice?

**Table 4 ijerph-19-01563-t004:** Demographic and clinical features of each patient rehabilitated with dental implants.

ID ^1^	Sex ^2^	Age	Subtype of OI ^3^	Smoker	Bisphosphonates ^4^	Ridge Augmentation ^4^	Antibiotics during Surgery ^4^	Sinus Floor Elevation ^4^	Number of Implants	Implant Characteristics	Two-Phase/One-Phase ^5^	Abutment Type ^6^	Follow-Up Time ^7^	Implant Survival
1 [20]	F	32	NA	NA	NA	Yes	NA	Yes	5	Straumann standard implants	Tp	r+od	48	5/5
2 [8]	M	61	IV	Yes	Yes	NA	Yes	No	3	MIS C1 3.75 × 10 /3.75 × 11.5	Tp	c	48	3/3
3 [18]	F	51	NA	NA	NA	NA	Yes	Yes	6	Regular-platform TiUnite Brånemark System Implants	Tp	r+od	48	6/6
4 [7]	F	64	I	NA	Yes	No	Yes	No	2	Straumann Standard Plus SLActive 3.3 mm × 10 mm	Tp	l+od	12	2/2
5 [19]	F	73	I	No	NA	NA	NA	NA	1	Nobel	Na	b	>120	1/1
6 [19]	M	52	Ib	Yes	NA	NA	NA	NA	5	AstraTech Tioblast 3.5 × 15/3.5 × 17	Tp	b	135	5/5
7 [19]	M	69	I	Yes	NA	NA	NA	NA	5	AstraTech Tioblast 4.0 × 15/3.5 × 11	Tp	b	60	5/5
8 [19]	M	49	IV	No	NA	NA	NA	NA	1	AstraTech Tioblast 4.5 × 13	Na	c	79	1/1
9 [19]	F	58	IV	Yes	NA	NA	NA	NA	6	AstraTech Tioblast 3.5 × 13/4.0 × 13/3.5 × 15	Tp	c+od	83	6/6
10 [19]	F	52	III	Yes	NA	NA	NA	NA	7	AstraTech Osseospeed 3.5 × 9/3.5 × 11 NobelDirect 3.0 × 15/3.0 × 10	Mixed	c+od	29–57	5/7
11 [19]	M	75	I	No	NA	NA	NA	NA	7	Straumann 4.1 × 12 AstraTech Osseospeed 3.5 × 13	Mixed	c+od	11–22	7/7
12 [19]	F	65	Ib	No	NA	NA	NA	NA	2	AstraTech Osseospeed 3.5 × 13/4.0 × 13	Tp	c	23	2/2
13 [9,19]	M	58	Ib	Yes	NA	NA	NA	NA	5	AstraTech Osseospeed 4.0 × 13	Tp	c	103–109	5/5
14 [19]	M	20	III	No	NA	NA	NA	NA	1	Biomet 3i tapered 3.25 × 11	Op	c	22	1/1
15 [9,19]	M	39	Ib	No	NA	NA	NA	NA	3	AstraTech Osseospeed 3.5 × 13 Biomet 3i tapered 3.25 × 11	Tp	c	104–106	3/3
16 [9,19]	F	48	I	No	NA	NA	NA	NA	2	AstraTech Osseospeed 4.5 × 11/5.0 × 11	Tp	c	76–91	1/2
17 [9,19]	F	56	IV	Yes	Yes	NA	NA	NA	1	Straumann 4.1 × 8	Tp	c	94	1/1
18 [17]	F	43	III	NA	NA	Yes	NA	No	2	Paragon Screw-vent internal hexed implants	Tp	b	24	2/2
19 [22]	F	34	IV	NA	NA	Some implants	Yes	Some implants	11	Brånemark System Mk III Ti-Unite implants 3.75 × 15/3.75 × 10/3.75 × 11.5/4 × 15	Tp	r+od	24	11/11
20 [24]	M	34	IVb	NA	NA	No	NA	No	11	Brånemark titanium bone-tapped implants 13/15	Tp	r+od	108	10/11
21 [21]	M	53	I	NA	No	Some implants	NA	NA	10	NobelActive Implant 3.5 × 13/3.5 × 10/4.3 × 10/4.3 × 11/4.3 × 13/5 × 10	Op	b/c	13–40	10/10
22 [16]	F	30	III	NA	Yes	Yes	NA	No	4	Astra Tech OsseoSpeed 3.5 × 11	Tp	b	36	4/4
23 [23]	M	20	NA	NA	NA	Some implants	Yes	Some implants	16	3.5 × 10/3.5 × 13/4.0 × 10/4.0 × 13/4.0 × 15	Tp	b	65–86	13/16

^1^ The reference to the respective study is given in parentheses. ^2^ F—female; M—male. ^3^ Subtype according to the Sillence classification [3]; an additional diagnosis of dentinogenesis imperfecta is marked with a “b”; NA—not answered. ^4^ Detailed information can be found in the respective results. ^5^ Tp—two-phase procedure; Op—one-phase procedure with immediate loading; Mixed—combination of both one- and two-phase procedures for different implants. ^6^ r—ridge; c—crown; b—bridge; od—overdenture. ^7^ Follow-up time in months; range is reported when follow-up differs between implants.

**Table 5 ijerph-19-01563-t005:** Implant positions and respective survival/failure rates.

Implant Region	No. Implants (%)	Implant Survival (%)	Implant Failure (%)
17–14	18 (15.5)	16 (88.9)	2 (1.1)
13–23	26 (22.4)	26 (100)	0 (0)
24–28	22 (18.9)	21 (95.5)	1 (4.5)
38–34	17 (14.7)	15 (88.2)	2 (11.8)
33–43	19 (16.4)	18 (94.7)	1 (5.3)
44–48	14 (12.1)	13 (92.9)	1 (7.1)
Total	116 (100)	109 (94.0)	7 (6.0)

**Table 6 ijerph-19-01563-t006:** Implant outcome at time of follow-up.

	Mean	SD	Range
Implants per patient	5.0	±3.8	1–16
Time between implantation and loading (months)	8.9	±6.3	0–40
Follow-up time	59.1	±36.1	11–135
Radiological bone loss (mm)	0.8	±1.3	0–7
Time between surgery and failure of implants (months)	39.1	±29.4	3–78

**Table 7 ijerph-19-01563-t007:** Implant survival rates in different treatment procedures and biological properties.

	No. of Subjects	No. of Implants	No. of Failed Implants	Survival Rate
**Subtype**				
I	10	42	1 *	97.%
III	4	14	2	85.7%
IV	6	33	1	97.0%
NA	3	17	3 *	88.9%
**Bone Augmentation**				
**Ridge Augmentation**				
Autogenous bone used	5	22	1	95.5%
Allograft used	1	8	0	100%
No augmentation	2	26	2	92.3%
NA	15	60	4 **	93.4%
**Sinus Floor Elevation**				
Sinus floor elevation performed	4	19	1	94.7%
Not performed	9	55	6 **	89.1%
NA (or information missing on some implants)	10	42	0	100%
**Surgical Procedure**				
Immediately loaded	4	14	0	100%
Two-stage procedure	18	100	7 **	93.0%
NA	2	2	0	100%
**Abutment**				
Crown	10	24	1 *	95.8%
Bridge	7	31	2 *	93.5%
Crown + Overdenture	3	17	0	100%
Locator + Overdenture	1	2	0	100%
Ridge + Overdenture	4	32	0	100%
NA	1	7	1	85.7%
Failed before loading	2	3	3	

NA—not answered; *—each “*” indicates one implant lost due to mechanical complications.

## Data Availability

The data are available upon request.

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
