# Peer review of "Dental Implants in People with Osteogenesis Imperfecta: A Systematic Review"

_ijerph, 2022, doi:10.3390/ijerph19031563_

Round 1

Reviewer 1 Report

First of all, I would like to congratulate you because I know the effort required to write a scientific paper.

But I have many things I should correct or change. 

You write in present and past many times; it is important to be in the same form (for example, in the abstract; line 14 (adresses), but in line 16 you write "were"). Please correct it in all the paper.

In line 47, I find important for readers to know how frequent agenesis are.

I would like to have more information about inclusion criteria (2.3; line 87) to better understand the selection of the articles.

In line 111, you talk about bone augmentation and about sinus lift; do not you think sinus lift is a bone augmentation procedure? you should correct it, because sinus lift augmentation is a surgical procedure to regenerate bone in the posterior maxilla...

In line 131, you analyze bone loss with x-ray, but, it is important to specify which type of x-ray.

When writing a scientific paper, it is more appropriate to be impersonal, so my advise is to delete we, our... (for example line 142, 144, 148, 180), and to change " we decided" and write, for example: "it was decided..."

In line 146 (methods) you write "the authors propose", it is important to write it in past form, and it is not necessary for readers to read the authors...be impersonal.

When talking about results, in Table 2 you can check implant number, but you speak about it in 3.9, I find it more interesting when you talk about patient number...

About biphosphonates, I find mandatory to know if cases where autogenous bone was harvested (iliac crest, mandibular ramus) were taking them, or not.

I do not understand when you exclude two implants due to implant fractures, do you consider implant survival when they are fractured? which are your survival rates criteria?  and in the articles included?

The last question is about the radiographic bone loss, how is it measured? 

Author Response

First of all, I would like to congratulate you because I know the effort required to write a scientific paper.

But I have many things I should correct or change. 

You write in present and past many times; it is important to be in the same form (for example, in the abstract; line 14 (adresses), but in line 16 you write "were"). Please correct it in all the paper.

We would like to thank you for all of your remarks. We checked the use of present and past throughout the paper and corrected all wrongfully used forms.

In line 47, I find important for readers to know how frequent agenesis are.

We added this information in lines 49-50.

I would like to have more information about inclusion criteria (2.3; line 87) to better understand the selection of the articles.

We clarified the inclusion criteria in lines 88-93.

In line 111, you talk about bone augmentation and about sinus lift; do not you think sinus lift is a bone augmentation procedure? you should correct it, because sinus lift augmentation is a surgical procedure to regenerate bone in the posterior maxilla...

Thank you for your comment, it was incorrectly written unclearly. Of course, we would consider sinus floor elevation to be a bone augmentation procedure. We realize that our wording suggests otherwise. When we refer to bone augmentation, ridge augmentation would have been the correct term. In both Methods (lines 115-116) and Results (lines 240-263), we have changed the wording and added two subheadings to our Bone Augmentation heading. 

In line 131, you analyze bone loss with x-ray, but, it is important to specify which type of x-ray.

Thank you for noticing, we specified in lines 129-130 that only single-toot radiographs or panoramic radiographs were used to analyze bone loss.

When writing a scientific paper, it is more appropriate to be impersonal, so my advise is to delete we, our... (for example line 142, 144, 148, 180), and to change " we decided" and write, for example: "it was decided..."

All uses of “we, our” and similar expressions have been changed to be impersonal.

In line 146 (methods) you write "the authors propose", it is important to write it in past form, and it is not necessary for readers to read the authors...be impersonal.

This sentence was rewritten because "the authors" meant the authors of the tool, Murad et al. In addition, the tense was adjusted accordingly (lines 151-153).

When talking about results, in Table 2 you can check implant number, but you speak about it in 3.9, I find it more interesting when you talk about patient number...

Thank you for the observation, we have added information about the number of patients affected. As we felt it was important to also have information on the number of implants placed, this information has been transferred with no change (lines 307-329).

About biphosphonates, I find mandatory to know if cases where autogenous bone was harvested (iliac crest, mandibular ramus) were taking them, or not.

We added this information in lines 254-256.

I do not understand when you exclude two implants due to implant fractures, do you consider implant survival when they are fractured? which are your survival rates criteria?  and in the articles included?

This was another case of unfortunate wording on our part. Of course, we do not consider the survival of an implant if it is fractured. We just wanted to emphasize that the focus of implant loss in this study should be on biological complications as they relate to the patient and possibly to the rare disease. Because mechanical complications can occur with any implant, regardless of the patient, it should be considered that these two failures cannot be attributed to the affected patient. Nevertheless, these implants failed and therefore should be counted as failed implants. The paragraph in the discussion has been reworded (line 446-461) and should now be clearer. 

The last question is about the radiographic bone loss, how is it measured? 

Detailed information on how radiographic bone loss was measured was provided in only two articles (lines 466-468). We added this information as a remark in the limitations because further studies should include detailed information on how radiographic bone loss was measured. Thank you.

Reviewer 2 Report

Through a systematic review of analyzed clinical cases, authors assessed possibilities of prosthetic rehabilitation with dental implants in patients with the rare disease, osteogenesis imperfecta (OI). From the objective reason of the rare disease, this review included a very small number of analyzed papers.

As it can be seen from the small number of published case reports and case series of patients with OI  presented in the literature, there is not enough clinical data on this topic or broader scientific knowledge, so this systematic review summarizes previous clinical experiences and in the same time opens a lot of new questions and possibilities.

With regard to the  particular characteristics of the craniofacial complex in this patients, in order to evaluate the improvement of any treatment in OI patients, and particularly surgical treatment with dental implants and to evaluate treatment outcome, data of the craniofacial complex as well as malocclusion, which becomes more prominent  with increased age, should be established. Characteristics of the craniofacial complex and type of OI in these patients could certainly affect the therapy and implant survival rate.  Therefore authors well conclude  that  providing more data in clinical studies, which include the assessment of clinical and radiological status of the jawbone, craniofacial findings, type of OI disease and therapy, should be taken  into consideration in every case. Taking into account such approach, this systematic review opens up the possibility for creating clinical guidelines to be established which patient with OI could be the proper candidate for implant-prosthetic rehabilitation.

Therefore, I recommend this systematic review  to be accepted.

Author Response

Through a systematic review of analyzed clinical cases, authors assessed possibilities of prosthetic rehabilitation with dental implants in patients with the rare disease, osteogenesis imperfecta (OI). From the objective reason of the rare disease, this review included a very small number of analyzed papers.

As it can be seen from the small number of published case reports and case series of patients with OI  presented in the literature, there is not enough clinical data on this topic or broader scientific knowledge, so this systematic review summarizes previous clinical experiences and in the same time opens a lot of new questions and possibilities.

With regard to the  particular characteristics of the craniofacial complex in this patients, in order to evaluate the improvement of any treatment in OI patients, and particularly surgical treatment with dental implants and to evaluate treatment outcome, data of the craniofacial complex as well as malocclusion, which becomes more prominent  with increased age, should be established. Characteristics of the craniofacial complex and type of OI in these patients could certainly affect the therapy and implant survival rate.  Therefore authors well conclude  that  providing more data in clinical studies, which include the assessment of clinical and radiological status of the jawbone, craniofacial findings, type of OI disease and therapy, should be taken  into consideration in every case. Taking into account such approach, this systematic review opens up the possibility for creating clinical guidelines to be established which patient with OI could be the proper candidate for implant-prosthetic rehabilitation.

Therefore, I recommend this systematic review  to be accepted.

Thank you for this review and your recommendation.

Reviewer 3 Report

One problem of the present study is that it takes into consideration the overall cases of implant restorations, even though the patients had very different treatment plans.

While some went through major bone augmentation procedures, others had just the basic implant surgery.

The same way, the different prosthetic approaches weren't taken into account; especially when dealing with a low quality bone, the single restorations can behave very different than a partial bridge or an all-on-x.

So, in order to be more specific, the authors should make a more realistic comparison between patients with similar treatment plans and biologic features.

Otherwise, too many factors can affect the results so that it cannot be draw an accurate conclusion.

Author Response

One problem of the present study is that it takes into consideration the overall cases of implant restorations, even though the patients had very different treatment plans.

While some went through major bone augmentation procedures, others had just the basic implant surgery.

The same way, the different prosthetic approaches weren't taken into account; especially when dealing with a low quality bone, the single restorations can behave very different than a partial bridge or an all-on-x.

So, in order to be more specific, the authors should make a more realistic comparison between patients with similar treatment plans and biologic features.

Otherwise, too many factors can affect the results so that it cannot be draw an accurate conclusion.

First of all, we would like to thank you for your comments. This are great remarks that we did not consider as carefully as we should have. Since this systematic review aimed to provide practitioners with a first systematic approach to implant survival rates in patients with osteogenesis imperfecta, we decided to include all cases of implants placed in these patients to maximize the small number of total implants. As you mentioned, this significantly increases the number of different treatment protocols and prosthetic approaches. To allow readers to compare the different surgical protocols and prosthetic approaches, we decided to add a table (line 350 and following) reporting the survival rate of different subgroups with information on the number of patients and implants. We have also added a paragraph to the limitation (lines 477-488) focusing on your comments so that the reader can take into account that this systematic review considers a very heterogeneous group. We believe that your comments have really contributed to the review and provide a more accurate view of the available data.

Reviewer 4 Report

Dear Sirs, thank you for the opportunity to review this paper. There are some small issues I would like to add:

  • please, add the description of PRISMA flow diagram in the text (with reference to it - fig. 1)
  • please add the information from table 4 in the main text - especially ones concerning implants used and which of these were stable or not
  • please, add information on exactly how biophosphonates influence the dentistry (eg. orthodontic treatment, especially in terms of pre-prosthetic restoration) in the discussion session - eg. lines 333 and furhter
  • please add in line 393 the information why do the Authors find the implant survival rate lower - a short explanation, that was mentioned before should be placed in here as well
  • in conclusion there should be a statement that the survival rate of implants is still really high and that they should be a method of choice when restoring dentition in OI patients.

In my oppinion, after that tiny, rather cosmetic changes, the paper should be published.

Author Response

Dear Sirs, thank you for the opportunity to review this paper. There are some small issues I would like to add:

  • please, add the description of PRISMA flow diagram in the text (with reference to it - fig. 1)

Thank you for you remarks. We added the description in lines 167-170.

  • please add the information from table 4 in the main text - especially ones concerning implants used and which of these were stable or not

A paragraph focusing on the characteristics of placed implants has been added in lines 282-305 with information on implants that failed.

  • please, add information on exactly how biophosphonates influence the dentistry (eg. orthodontic treatment, especially in terms of pre-prosthetic restoration) in the discussion session - eg. lines 333 and further

We added information on the influence of bisphosphonates in dentistry in lines 386-395.

  • please add in line 393 the information why do the Authors find the implant survival rate lower - a short explanation, that was mentioned before should be placed in here as well

Thank you for noticing, a short explanation was added in lines 459-461.

  • in conclusion there should be a statement that the survival rate of implants is still really high and that they should be a method of choice when restoring dentition in OI patients.

The statement in the conclusion was rewritten (lines 496-499) to stronger emphasize the implication of a high survival rate and that it should be a method of choice for patients with OI.

  • In my oppinion, after that tiny, rather cosmetic changes, the paper should be published.

We would like to thank you for all of your comments and your recommendation.

Round 2

Reviewer 1 Report

I think with the changes you have performed it is better

Reviewer 3 Report

The authors submitted an improved version of the manuscript. It can be published.